Identification and characterization of a novel zebrafish (Danio rerio) pentraxin–carbonic anhydrase

http://orcid.org/0000-0001-9698-278X Patrikainen Maarit S. 1
Tolvanen Martti E.E. 2 martti.tolvanen@utu.fi
http://orcid.org/0000-0002-6938-7835 Aspatwar Ashok 1 3
Barker Harlan R. 1
Ortutay Csaba 4
http://orcid.org/0000-0002-8446-4704 Jänis Janne 5
Laitaoja Mikko 5
http://orcid.org/0000-0002-9357-1480 Hytönen Vesa P. 1 3
Azizi Latifeh 1
Manandhar Prajwol 1 6
http://orcid.org/0000-0003-2601-8391 Jáger Edit 7
Vullo Daniela 8
Kukkurainen Sampo 1
http://orcid.org/0000-0003-0663-5843 Hilvo Mika 1 9
Supuran Claudiu T. 8
http://orcid.org/0000-0001-7323-8536 Parkkila Seppo 1 3
1 Faculty of Medicine and Life Sciences, University of Tampere , Tampere , Finland
2 Department of Future Technologies, University of Turku , Turku , Finland
3 Fimlab Ltd., Tampere University Hospital , Tampere , Finland
4 HiDucator Ltd. , Kangasala , Finland
5 Department of Chemistry, University of Eastern Finland , Joensuu , Finland
6 Center for Molecular Dynamics Nepal , Kathmandu , Nepal
7 Department of Epidemiology, Faculty of Health Sciences, Semmelweis University , Budapest , Hungary
8 Dipartimento Neurofarba, Sezione di Scienze Farmaceutiche e Nutraceutiche, Università degli Studi di Firenze , Florence , Italy
9 Zora Biosciences Ltd. , Espoo , Finland
Rojas Ana
Electronic publication date: 2017 Dec 7
Publication date: 2017
Volume: 5
Electronic Location ID: e4128
Received 2017 Aug 30; Accepted 2017 Nov 14
Copyright: © 2017 Patrikainen et al.
Copyright year: 2017
Copyright holder: Patrikainen et al.
License: This is an open access article distributed under the terms of the Creative Commons Attribution License, which permits unrestricted use, distribution, reproduction and adaptation in any medium and for any purpose provided that it is properly attributed. For attribution, the original author(s), title, publication source (PeerJ) and either DOI or URL of the article must be cited.
License URL: https://creativecommons.org/licenses/by/4.0/

Keywords: Phylogeny, Protein modeling, Pentraxin, Zebrafish, Carbonic anhydrase VI, Carbonic anhydrase, Mass spectrometry, Innate immunity, Knockdown

Funding: Jane & Aatos Erkko Foundation Sigrid Jusélius Foundation Tampere Tuberculosis Foundation Finnish Cultural Foundation Academy of Finland This work was supported by grants from the Jane & Aatos Erkko Foundation (Seppo Parkkila), the Sigrid Jusélius Foundation (Seppo Parkkila), the Tampere Tuberculosis Foundation (Seppo Parkkila), the Finnish Cultural Foundation (Harlan R. Barker), and the Academy of Finland (Seppo Parkkila, Vesa P. Hytönen). There was no additional external funding received for this study. The funders had no role in study design, data collection and analysis, decision to publish, or preparation of the manuscript.

==============================
Background

Carbonic anhydrases (CAs) are ubiquitous, essential enzymes which catalyze the conversion of carbon dioxide and water to bicarbonate and H+ ions. Vertebrate genomes generally contain gene loci for 15–21 different CA isoforms, three of which are enzymatically inactive. CA VI is the only secretory protein of the enzymatically active isoforms. We discovered that non-mammalian CA VI contains a C-terminal pentraxin (PTX) domain, a novel combination for both CAs and PTXs.

Methods

We isolated and sequenced zebrafish (Danio rerio) CA VI cDNA, complete with the sequence coding for the PTX domain, and produced the recombinant CA VI–PTX protein. Enzymatic activity and kinetic parameters were measured with a stopped-flow instrument. Mass spectrometry, analytical gel filtration and dynamic light scattering were used for biophysical characterization. Sequence analyses and Bayesian phylogenetics were used in generating hypotheses of protein structure and CA VI gene evolution. A CA VI–PTX antiserum was produced, and the expression of CA VI protein was studied by immunohistochemistry. A knock-down zebrafish model was constructed, and larvae were observed up to five days post-fertilization (dpf). The expression of ca6 mRNA was quantitated by qRT-PCR in different developmental times in morphant and wild-type larvae and in different adult fish tissues. Finally, the swimming behavior of the morphant fish was compared to that of wild-type fish.

Results

The recombinant enzyme has a very high carbonate dehydratase activity. Sequencing confirms a 530-residue protein identical to one of the predicted proteins in the Ensembl database (ensembl.org). The protein is pentameric in solution, as studied by gel filtration and light scattering, presumably joined by the PTX domains. Mass spectrometry confirms the predicted signal peptide cleavage and disulfides, and N-glycosylation in two of the four observed glycosylation motifs. Molecular modeling of the pentamer is consistent with the modifications observed in mass spectrometry. Phylogenetics and sequence analyses provide a consistent hypothesis of the evolutionary history of domains associated with CA VI in mammals and non-mammals. Briefly, the evidence suggests that ancestral CA VI was a transmembrane protein, the exon coding for the cytoplasmic domain was replaced by one coding for PTX domain, and finally, in the therian lineage, the PTX-coding exon was lost. We knocked down CA VI expression in zebrafish embryos with antisense morpholino oligonucleotides, resulting in phenotype features of decreased buoyancy and swim bladder deflation in 4 dpf larvae.

Discussion

These findings provide novel insights into the evolution, structure, and function of this unique CA form.

Introduction

Carbonic anhydrase VI (CA VI) is the only secretory CA enzyme in mammals. In its very first reporting, Henkin et al. (1975) described a novel protein, gustin, from human saliva, which was later shown to be CA VI (Thatcher et al., 1998). This protein was first described as a CA enzyme by Fernley, Wright & Coghlan (1979), who identified a novel high molecular weight (MW) form of CA in the sheep parotid gland and saliva. The first immunohistochemical studies on human CA VI indicated that it is highly expressed in the serous acinar cells of the parotid and submandibular glands (Parkkila et al., 1990). It is one of the major protein constituents of human saliva (Parkkila et al., 1993), and also found in human and rat milk (Karhumaa et al., 2001).

The physiological role of CA VI has remained unclear, even though it was discovered three decades ago. Henkin’s group linked gustin (CA VI) to the regulation of taste function (Shatzman & Henkin, 1981). Expression of CA VI in the von Ebner’s glands implicate CA VI in the paracrine modulation of taste function and TRC apoptosis (Leinonen et al., 2001). Various studies have later shown a link between bitter taste perception and CA VI (Melis et al., 2013; Patrikainen et al., 2014). Two studies have shown a link between CA VI and immunological function in mouse and human. First, Car6−/− mice have a greater number of lymphoid follicles in the small intestinal Peyer’s patches, suggesting an immunological phenotype (Pan et al., 2011). Second, the analysis of gene expression in the trachea and lung of Car6−/− mice showed alterations in biological processes such as antigen transfer to mucosal-associated lymphoid tissue (Patrikainen et al., 2016).

Innate immune systems, based on pattern recognition, exist in some form in all metazoan organisms (Medzhitov, 2007). The pattern-recognition molecules (PRMs) recognize conserved structures on the surface of pathogens and activate the innate immune response. Pentraxins (PTXs) are a superfamily of fluid phase PRMs conserved in evolution and characterized by a cyclic multimeric structure with a regulatory role in inflammation (Bottazzi et al., 2016). They contain a characteristic ∼200-residue-long domain at their C-terminus. Based on their primary subunit structures PTXs are divided into short PTXs and long PTXs. Short PTXs are classically represented by C-reactive protein (a.k.a. CRP, pentraxin-1, PTX-1) and serum amyloid P (a.k.a. APCS, SAP, pentraxin-2, PTX-2), whereas long PTXs comprise pentraxin-3 (PTX3), neuronal PTXs, and others (Garlanda et al., 2005).

We noted the presence of an additional PTX domain in some non-mammalian CA6 gene predictions in 2007, but did not follow up this observation at that time. More recently, with more non-mammalian genomes available, we realized that the PTX domain is present in non-mammalian CA VI too consistently to be an annotation artifact, which inspired this study. We used zebrafish (Danio rerio) as a vertebrate model organism for functional and structural characterization of the PTX-associated CA VI.

Materials and Methods

Sequence conservation

In order to compare conservation in the CA and PTX domains of CA VI–PTX proteins, non-mammalian CA VI sequences were retrieved from NCBI (NCBI Resource Coordinators, 2016) nr protein database as of December 5, 2015, using BLASTP (https://blast.ncbi.nlm.nih.gov/Blast.cgi?PAGE=Proteins) (Altschul et al., 1990), with human CA VI (ENSP00000366662 from Ensembl (Flicek et al., 2012) as query sequence, taxonomically filtered for non-mammalian vertebrates. Full-length or nearly full-length CA VI–PTX sequences were seen at extremely low e values, not higher than 2 × 10−80, indicating very high similarity; CA VI sequence fragments were seen at e values from 10−79 to 10−71; and the remaining matches, at e values of 10−68 and higher, were annotated as other CA isoforms and did not contain a PTX domain. Sequences with an e value of 2 × 10−80 or lower were taken for further quality control. We discarded sequences shorter than 485 residues and any with non-specific “X” characters. Furthermore, we rejected sequences with unaligned, unique insertions of at least 20 residues at exon boundaries, which we assume to be introns mispredicted as coding sequence. Likewise, sequences containing gaps in the alignment between exon boundaries were interpreted to miss data for internal exons and were discarded. Thus, all sequences which were incomplete in the CA domain were discarded, but sequences devoid of the signal peptide region were still kept. The final sequence set contained 78 sequences from 75 species, (sequence accession numbers shown in Fig. S1 and in Data S1). After inspection of the multiple sequence alignment, four sequences were edited for a more plausible initiation site (Table 1) deemed to be at the conserved M at the start of the signal peptide region. All sequences had complete PTX domains. Sequences were aligned with Clustal Omega (Sievers et al., 2011). In order to calculate conserved positions in each domain, the CA domain was defined to correspond to residues 24–280 in zebrafish CA VI (UniProt annotation in E9QB97_DANRE), and the PTX domain was defined as residues 317–518 (InterProScan at http://www.ebi.ac.uk/interpro/sequence-search (Jones et al., 2014), match to profile SM00159, PTX).

Table 1 Suggested corrections for predicted translation start sites.

RefSeq ID	Name	Organism	Number of N-terminal residues removed	
XP_010721064.1	PREDICTED: carbonic anhydrase 6	Meleagris gallopavo	110	
XP_005057921.1	PREDICTED: carbonic anhydrase 6	Ficedula albicollis	17	
XP_002187446.1	PREDICTED: carbonic anhydrase 6	Taeniopygia guttata	6	
XP_005143337.1	PREDICTED: carbonic anhydrase 6	Melopsittacus undulatus	31	
Note:

The following database entries have N-terminal extensions which we assume mispredicted. We have shortened these sequences to start at a conserved initiating Met residues for use in this study.

Phylogenetic analyses

For the tree in Fig. 1, cDNA sequences and their protein translations were collected from the Ensembl database (release 67) for CAs 6, 9, 12, and 14, from selected species. Protein sequences were aligned with Clustal Omega. Codon aligned cDNA sequences were produced in the PAL2NAL web server v. 14 (http://www.bork.embl.de/pal2nal/) (Suyama, Torrents & Bork, 2006) using the protein alignment as a guide (protein alignment: Data S2; final codon alignment: Data S3). For the tree in Fig. S2, a second alignment was similarly made using catalytic domains of CA VI sequences only (protein alignment: Data S4; final codon alignment: Data S5). For the tree in Fig. 2, we made a third alignment of CA VI-associated PTX domains from selected species and human PTXs (codon aligned sequences: Data S6). The resulting codon (DNA) alignments and the program MrBayes v 3.2 (Ronquist et al., 2012) were used to estimate the phylogeny of the sequences by Bayesian inference. Bayesian estimation was run for at least 10,000 generations, with flat a priori distribution of base frequencies, substitution rates, proportion of invariable sites, and gamma shape parameter. The 50% majority rule consensus trees were saved and visualized using the APE R package (Paradis, Claude & Strimmer, 2004).

Figure 1 Bayesian phylogenetic tree of CA VI, CA IX, CA XII, and CA XIV.

Analysis of protein alignment guided DNA alignments as detailed in “Materials and Methods.” Sidebars indicate the groups of isoforms. The CA VI subtree with more species is shown in Fig. S2.

Figure 2 Bayesian phylogenetic tree of pentraxin domains.

Analysis of protein alignment guided DNA alignments as detailed in “Materials and Methods.” Sidebars indicate PTX domains extracted from non-mammalian CA VI sequences (bottom) and groups of human pentraxins.

Run lengths, relevant parameters at the end of run, and rooting of the trees were as follows. For the first tree, the average standard deviation of split frequencies after 10,000 generations was 5.2 × 10−2 when the analysis was stopped. The arithmetic mean of the estimated marginal likelihoods for runs sampled was −17175.07. Drosophila melanogaster CAH1 sequence was used as an outgroup to root the tree. For the second tree, the average standard deviation of split frequencies after 20,000 generations was 1.2 × 10−1 when the analysis was stopped. The arithmetic mean of the estimated marginal likelihoods for runs sampled was −10596.5. Fish sequences were used as an outgroup to root the consensus tree. Branching points with lower than 50% consensus in the mammal branch are collapsed. Finally, for the third tree, the average standard deviation of split frequencies after 10,000 generations was 8.1 × 10−2 when the analysis was stopped. The arithmetic mean of the estimated marginal likelihoods for runs sampled was −10319.1.

BlastN search in platypus genome

In order to see if the orphan fragment Contig22468 of platypus genome (which contains the exon coding for a “CA VI-type” PTX domain) would have been somehow missed in the genome assembly, we performed a BlastN search in Ensembl (http://www.ensembl.org/Homo_sapiens/Tools/Blast?db=core). BlastN was run against the platypus genome with the full 11,311 nt sequence of supercontig:OANA5:Contig22468 as query sequence.

Exon length comparisons

Exon data was retrieved from Ensembl. Lengths of the exons that follow those coding for the CA domain were noted for Ensembl transcripts for human CA6 (ENST00000377443), CA9 (ENST00000378357), CA12 (ENST00000178638 and ENST00000344366), and CA14 (ENST00000369111), and zebrafish ca6 (ENSDART00000132733). Similarly, lengths of the exons preceding the PTX domain exon were noted for Ensembl transcripts of human CRP (ENST00000255030), APCS (SAP, ENST00000255040).

Amphipathic helix prediction

A study of the region between the CA and PTX domains in the alignment of CA VI protein sequences showed little conservation except for five sites with hydrophobic residues spaced three or four residues apart, with mostly polar residues between them, suggestive of an amphipathic alpha helix. The subsequences from 287 to 303 and from 293 to 309 for human and zebrafish CA VI, respectively, were visualized as helical wheel diagrams, or end projections of a hypothetical alpha helix of 17 residues, using the PepWheel program of the EMBOSS suite (http://www.bioinformatics.nl/cgi-bin/emboss/help/pepwheel) (Rice, Longden & Bleasby, 2000).

Construction of recombinant baculoviruses

The 1,593 bp zebrafish ca6 sequence encoding the full-length, PTX-containing CA VI polypeptide (CA VI–PTX) was amplified by PCR using the forward primer 5′-ATGGAGCAGCTGACTCTAGTC-3′ and reverse primer 5′-TTTCTCTGTTTCTCTATTATTATTAT-3′. PCR conditions consisted of an initial denaturation step at 98 °C for 30 s followed by 35 cycles at 98 °C for 10 s (denaturation), 55 °C for 30 s (annealing), and 72 °C for 25 s (elongation). The final extension step was carried out at 72 °C for 5 min. The expression construct was optimized for protein production in Spodoptera frugiperda insect cells (Sf9) by inserting into second round PCR primers restriction sites for BamHI and SalI plus sequences coding for C-terminal histidine tag for protein purification and a thrombin cleavage site for tag removal. The second round of PCR was carried out using the protocol described above, except that the temperature for annealing was 62 °C, and final extension step was carried out at 74 °C for 7 min. The baculoviral genomes encoding CA VI recombinant proteins were generated according to the Bac-to-Bac Baculovirus Expression System instructions (Invitrogen, Camarillo, CA, USA). The recombinant protein insert was sequenced using ABI PRISM BigDye® Terminator v3.1 Cycle Sequencing kit (Applied Biosystems, Inc., Foster City, CA, USA.) and pFASTBac primers (forward: 5′-AATGATAACCATCTGGCA-3′ and reverse: 5′-GGTATGGCTGATTATGAT-3′) in order to obtain the full-length insert sequence. The PCR conditions consisted of 35 cycles at 96 °C for 10 s (denaturation), 50 °C for 5 s (annealing), and 55 °C for 4 min (elongation) with final extension at 37 °C for 5 min.

Production and purification of recombinant CA VI–PTX

The Sf9 insect cells (Invitrogen) were maintained in HyQ SFX-Insect serum-free cell culture medium (HyClone, Logan, UT, USA). The cells were centrifuged (2,000×g, at 20 °C, for 5 min) 72 h after infection, and the medium was collected. Purification was performed with the Probond® Purification System (Invitrogen) under native binding conditions with wash and elution buffers made according to the manufacturer’s instructions. Purity of the protein was checked and the MW of the recombinant protein was determined by running a 10% SDS-PAGE (sodium dodecyl sulfate-polyacrylamide gel electrophoresis) under reducing conditions. The size of the protein was determined using Precision Plus Protein™ Standards Dual Color (Bio-Rad Laboratories, Inc., Hercules, CA, USA) and MW marker and bands were visualized using the Colloidal Blue Staining Kit™ (Invitrogen).

CA activity/inhibition assay

An Applied Photophysics stopped-flow instrument was used for assaying the CA catalyzed CO2 hydration activity (Khalifah, 1971). The method was exactly as described previously (Berrino et al., 2017) except that the inhibitor dilutions were done up to 0.5 nM.

Light scattering experiments

Molecular weight determination of zebrafish CA VI–PTX was performed using a Malvern Zetasizer μV instrument (Malvern Instruments Ltd., Worcestershire, UK) running static light scattering (SLS) and dynamic light scattering (DLS) methods. Analysis was performed using a liquid chromatography instrument (CBM-20A, Shimadzu Corporation, Kyoto, Japan) equipped with autosampler (SIL-20A), UV–VIS (SPD-20A) and fluorescence detector (RF-20Axs). Data were processed using Lab Solution Version 5.51 (Shimadzu Corporation) and OmniSec 4.7 (Malvern Instruments Ltd., Worcestershire, UK) softwares. A sample of the protein (50 μg) was injected on a Superdex 200 5/150 column (GE Healthcare, Uppsala, Sweden) equilibrated with 50 mM NaH3PO4, 500 mM NaCl pH 8 buffer. Runs were performed with flow rate of 0.1 ml/min at 20 °C using a thermostated cabin. The MW of the zebrafish CA VI–PTX was determined either by using a standard curve based on MW standard proteins (SEC analysis; CA 29 kDa, alcohol dehydrogenase 150 kDa, β-amylase 200 kDa, BSA 66 kDa, Sigma-Aldrich, Inc., St. Louis, MO, USA) or by calibrating the light scattering detector using the monomeric peak of BSA and light-scattering intensity (SLS).

Sample preparation for mass spectrometry

Prior to the mass spectrometric measurements, the CA VI–PTX sample was buffer-exchanged to 10 mM ammonium acetate (pH 7.5) buffer using Sephadex G-25 M (PD-10) desalting columns (GE Healthcare, Gillingham, UK). Ten 1 ml fractions were collected, and the fractions containing protein were concentrated using Amicon Ultra (5 kDa cut-off) centrifugal filter devices (Merck Millipore, Darmstadt, Germany). Finally, protein concentrations were determined by UV-absorbance at 280 nm, using a sequence-derived extinction coefficient 99,155 M−1 cm−1, calculated by ProtParam at http://web.expasy.org/protparam/ (Gasteiger et al., 2003). Intact protein mass analysis was performed by diluting the sample to the desired protein concentration with acetonitrile (MeCN), containing 1% of acetic acid (HOAc). Alternatively, CA VI–PTX was digested with trypsin. Briefly, an aliquot of the CA VI–PTX sample was mixed with a sequencing grade modified trypsin (Promega, Madison, WI, USA) (3 mg/ml in water) to obtain a 1:20 (w/w) protease-to-protein ratio. The digest sample was incubated at 37 °C for 1 h, and subsequently diluted to approximately 10 μM with MeCN containing 1% HOAc.

Mass measurements and data analysis

All experiments were performed on a 12-T Bruker Solarix-XR FT-ICR mass spectrometer (Bruker Daltonik GmbH, Bremen, Germany), equipped with an Apollo-II electrospray ionization (ESI) source and a dynamically harmonized ParaCell ICR-cell. All protein/peptide samples were directly infused into the ESI source at a flow rate of 1.5 μL/min. The ESI-generated ions were externally accumulated in the hexapole collision cell for 1 s, and transferred to the ICR cell for trapping, excitation and detection. For each mass spectrum, a total of 300 time-domain transients (1 MWord each) were co-added, and zero-filled once to obtain final 2 MWord broadband data. For collision-induced dissociation tandem mass spectrometry (CID-MS/MS) experiments, the precursor ions of interest were mass-selected in a quadrupole and fragmented in the collision cell by increasing the collision voltage to the appropriate value. The mass spectra were externally calibrated with ESI-L Low concentration tuning mix (Part no G1969-85000; Agilent Technologies, Santa Clara, CA, USA). The instrument was operated and the data were acquired by using Bruker ftmsControl 2.0 software. The mass spectra were subsequently transferred to Bruker DataAnalysis 4.4 software for further processing. Spectral de-isotoping and charge-state deconvolution (to obtain monoisotopic peptide masses) was accomplished with a Bruker SNAP2 peak-picking module. The obtained mass lists were then uploaded to GPMAW 10.0 software (Lighthouse Data, Odense, Denmark) for tryptic peptide identification. Only specific tryptic peptides were searched with a maximum mass error of 5 ppm. The glycosylation sites in CA VI–PTX were identified by incorporating typical high-mannose/complex glycans with a various number of residues into the four putative N-glycosylation sites and searched against the obtained mass lists.

Homology modeling of zebrafish CA VI

We built a 3D model of zebrafish CA VI starting from PDB 3FE4, human CA VI (Pilka et al., 2012); and 4AVS, human SAP component (Kolstoe et al., 2014), as templates for the CA domain and PTX domain, respectively. The CA template includes residues 32–280 of human CA VI, missing 14 residues in the N-terminus of the mature protein, and 28 residues in the C-terminus. Briefly, the predicted amphipathic helix (APH) region of human CA VI (287–303) and four additional residues (283–286) were modeled as an alpha helix, and the helix was subsequently docked to the C-terminal face of 3FE4. This extended model of human CA VI was used as a template in homology modeling the CA domain plus APH of zebrafish CA VI. The model of the PTX domain of zebrafish CA VI was docked to the model of CA+APH domains. Finally, a pentameric model of CA VI–PTX was created by superimposing five copies of the monomer model over each monomer in the pentameric SAP structure (PDB 4AVS).

The C-terminal alpha helix was generated automatically ab initio for the region predicted to form an APH when the full sequence of human CA VI was given as a modeling target to I-TASSER 4.0 (Roy, Kucukural & Zhang, 2010). The helix was separated from the model and subsequently docked to 3FE4, using the HADDOCK 2.1 server (http://haddock.science.uu.nl/) (de Vries et al., 2007). A list of potential interface residues in 3FE4 were predicted through CPORT (http://milou.science.uu.nl/services/CPORT/) (de Vries & Bonvin, 2011), and only those lying on the C-terminal face of 3FE4 were set as interacting residues in docking. For the APH, the residues located on the hydrophobic side were chosen as interacting residues. The resulting model from this docking step was used as a template to model the full CA domain of zebrafish CA VI at the MODELLER server (Webb & Sali, 2016) using UCSF Chimera v. 1.10 (Pettersen et al., 2004) as the interface program. The PTX domain of zebrafish CA VI (residues 317–530) was also modeled by MODELLER, with 4AVS as template. The two partial models of zebrafish CA VI were joined by docking with HADDOCK, again predicting interacting residues with CPORT. Most of the predicted interacting residues were located near the C-terminus of the CA domain and near the N-terminus of the PTX domain, and the residues in these regions were chosen as active residues in docking. The structural superimpositions of the PTX domains of the CA VI–PTX monomer on the SAP pentamer (4AVS) were carried out with the MatchMaker tool in UCSF Chimera to generate the final pentamer model.

Zebrafish maintenance and ethical permissions

Wild-type zebrafish of the AB strain were maintained at 28.5 °C under standard conditions (Westerfield, 2007). We express the embryonic ages in hours post-fertilization (hpf) and days post-fertilization (dpf). Embryos/larvae were collected from the breeder tanks with a sieve and rinsed with embryonic medium (Sarsted, Nümbrecht, Germany) into Petri dishes. Embryos/larvae were kept in Petri dishes in embryonic medium supplemented with 1-phenyl-2-thiourea (Sigma-Aldrich) at 28.5 °C until they were used in experiments. The maximum number of larvae on a 9 cm diameter Petri dish was 50. Embryonic medium contained 5 mM NaCl, 0.17 mM KCl, 0.33 mM CaCl2, 0.33 mM MgSO4 and 10–5% methylene blue (Sigma-Aldrich). Zebrafish housing and care in the Zebrafish facility of the University of Tampere have been approved by the National Animal Experiment Board of Finland, administered through the Provincial Government of Western Finland, Province Social and Health Department Tampere Regional Service Unit (permit # LSLH-2007-7254/Ym-23). Using five-day old zebrafish as a model organism requires no specific ethical permission, neither does studying tissues collected from euthanized adult fish.

Morpholino injections of zebrafish embryos

Knockdown of ca6 was carried out using two different antisense morpholino oligonucleotides (MOs) (GeneTools LLC, Philomath, OR, USA): one translation-blocking (MO1 5′-CTGCCTGTGCTCTGAACTGTTTCTC-3′) and the other splicing-blocking, to target intron–exon boundary before exon 9 (MO2 5′-GCTTGCCTTGAGAAGGAAAGATCAT). The random control (RC) MOs (5′-CCTCTTACCTCAGTTACAATTTATA-3′) were used as control MOs. The supplied MOs were re-suspended in sterile water at 1 mM stock concentration. Immediately prior to injection, ca6-MOs were diluted to the intended concentration of 125 μM. In order to monitor injection efficiency, 0.2% dextran rhodamine B and 0.1% phenol red (final concentrations; Sigma, Poole, UK) were included in the solution, and the final KCl concentration was adjusted to 1 M. About 1 nl of antisense MO solution was injected into the yolk of approximately 500 one- to two-cell stage embryos, without randomization. The MO-injected embryos were screened for the presence of fluorescence after 24 h to select the true ca6 morphants using Lumar V1.1 fluorescence stereomicroscope (Carl Zeiss MicroImaging GmbH, Göttingen, Germany) and AxioVision software version 4.9. The non-fluorescent embryos were eliminated.

Microscopy and live image analysis of zebrafish phenotypes

Gross phenotypic appearance was analyzed by light-field microscopy. For each experiment, typically 10–20 ca6-MO-injected larvae were screened with a similar number of matched controls. Larvae were first euthanized using 0.05% tricaine (Sigma-Aldrich) in embryo medium and embedded in 17% high MW methyl cellulose in 15 × 30 mm transparent polypropylene Petri dish for taking images of the developing embryos/larvae from 1 to 5 dpf using Zeiss Stereo Microscope (Carl Zeiss MicroImaging GmbH, Göttingen, Germany) with NeoLumar S 1.5× Objective (Carl Zeiss MicroImaging GmbH, Göttingen, Germany). The images were analyzed with AxioVision software version 4.9. and scale bars were inserted. Images were cropped and assembled into composite images.

Isolation of total RNA and synthesis of cDNA

Total RNA was isolated at different stages of development, from 0 to 168 hpf whole embryos/larvae, and from different organs of the adult zebrafish. Total RNA was isolated from 30 μg samples using the RNeasy® Mini kit (Qiagen, Hilden, Germany) by following the manufacturer’s instructions. The concentration and purity of total RNA were determined using a Nanodrop UV/VIS Spectrophotometer at 260 and 280 nm. Reverse transcriptase PCR was performed using 0.1–5 μg of total RNA to synthesize the first strand cDNA using First Strand cDNA Synthesis kit (High-Capacity cDNA Reverse Transcription Kits; Applied Biosystems, Foster City, CA, USA) with random primers and M-MuLV reverse transcriptase according to the protocol recommended by the manufacturer.

Quantitative real-time PCR

Quantitative real-time PCR (qRT-PCR) primers were designed based on the complete cDNA sequence taken from Ensembl (ENSDART00000057097), using the program Primer Express® Software v2.0 (Applied Biosystems) (forward primer 5′-CAAACATTTATTTGCCAGCACTCC-3′ and reverse primer 5′-TATGTCCAATAATCTCCATCTACTCC-3′). qRT-PCR was performed using the SYBR Green PCR Master Mix Kit in an ABI PRISM 7000 Detection System™ according to the manufacturer’s instructions (Applied Biosystems). The PCR conditions consisted of an initial denaturation step at 95 °C for 10 min followed by 40 cycles at 95 °C for 15 s (denaturation) and 60 °C for 1 min (elongation). The data were analyzed using the ABI PRISM 7000 SDS™ software (Applied Biosystems). Every PCR was performed in a total reaction volume of 15 μl containing 2 μl of first strand cDNA (20 ng cDNA), 1 × Power SYBR green PCR Master Mix™ (Applied Biosystems, Foster City, CA, USA), and 0.5 μM of each primer. We performed these experiments in duplicate and with sample duplicates. The results of ca6 gene expression were normalized using zebrafish housekeeping gene gapdh as internal control. The final results are given as relative expression values, calculated according to the Pfaffl’s equation (Pfaffl, 2001).

Preparation of zebrafish tissues

The adult zebrafish were euthanized by keeping them in 1% tricaine on ice for more than 10 min followed by decapitation. Different organs were harvested under the microscope and immediately transferred them to 1.5 ml microcentrifuge tube containing RNAlater® (Ambion, Austin, TX, USA) and were stored at −20 °C until further analysis. Simultaneously tissues for immunohistochemical analysis were harvested and immediately fixed with 4% PFA for 24 h at 4 °C. Tissues were transferred to 20% sucrose in PBS and stored at 4 °C until embedding them in Tissue-Tek® O.C.T.™ Compound (Sakura Finetek Europe B.V., Alphen aan den Rijn, The Netherlands). Embedded tissue samples were stored at −20 °C until further analysis.

Antibody testing

Antibody against zebrafish CA VI–PTX was manufactured by Innovagen AB (Innovagen AB, Lund, Sweden) according to their standard immunization schedule, with boosters at 14, 28, 49, and 70 days. Pre-immune serum and three samples of polyclonal antiserum were tested using dot blotting. Bio-Dot® Microfiltration Apparatus (BioRad Laboratories, Inc., Hercules, CA, USA) was used to attach 500 ng of produced and purified native zebrafish CA VI–PTX protein to PROTRAN® nitrocellulose (NC) transfer membrane (Schleicher & Schuell GmbH, Dassel, Germany) according to manufacturer’s instructions. Prior to staining, non-specific binding of the primary antibody was prevented using diluted colostrum (1:10 in Tris-Buffered Saline with Tween 20 [TBST]) as a blocking agent for 30 min. Pre-immune serum, bleed 1 (day 41), bleed 2 (day 62), and bleed 3 (day 83) of polyclonal rabbit anti-zebrafish CAVI-PTX (Innovagen AB, Lund, Sweden), diluted 1:100 in TBST, were added to NC strips which were incubated at room temperature for 1 h. Donkey anti-rabbit IgG, horseradish peroxidase linked whole antibody (Amersham Biosciences, GE Healthcare Life Sciences, Little Chalfont, UK) diluted 1:25,000 in TBST was used as secondary antibody. Washing steps were carried out using TBST. Staining was carried out using ImmPACTTM DAB Peroxidase Substrate Kit (Vector Laboratories, Inc., Burlingame, CA, USA). The testing showed that bleed 1 and bleed 2 have a strong reactivity against zebrafish CA VI–PTX (Figs. S3B and S3C). Antiserum of bleed 2 was used in further experiments.

Immunohistochemistry of zebrafish tissues

The Tissue-Tek® O.C.T.™ Compound-embedded samples were cut into 10 μm sections using cryotome and prior to staining, the sections were attached to the glass slide by incubating at 37 °C overnight. Staining procedure of tissue samples was carried out as described above. Alexa Fluor® goat anti-rabbit IgG 1:1,000 (Life Technologies, Carlsbad, CA, USA) was used as a secondary antibody, and sections were mounted with Vectashield Hard Set Mounting Medium with nuclear dye DAPI (4′,6-diamidino-2-phenylindole, Vector Laboratories Inc., Burlingame, CA, USA). The sections were photographed using Zeiss LSM780 Laser Scanning Confocal Microscope with Zeiss Cell Observer.Z1 microscope, Plan-Apochromat 40×/1.4 (oil) objective, with pulsed diode laser 405 nm and multiline Argon laser: 488 nm, and Quasar spectral GaAsP PMT array detector (Carl Zeiss Microscopy GmbH, Göttingen, Germany). Images were analysed with Zeiss ZEN2Lite.

Behavioral analysis of 4 and 5 dpf ca6 knockdown zebrafish larvae

Larvae were tested for behavioral consequences due to ca6 knockdown by measuring swimming pattern at 4 and 5 dpf. The ca6 knockdown larvae and two controls, namely uninjected wild-type and RC MO-injected, were raised in embryo medium. Larvae (approximately 10/flask) were placed in a 23 × 43 × 45 mm TC Flask T25 (Sarstedt AG & Co., Nümbrecht, Germany) containing 40 ml embryo medium at 3 dpf and allowed to acclimate to the flask for 24 h at 28.5 °C standard conditions. At 4 and 5 dpf their swimming patterns were observed by a 1 min video recording, with a printed 1 × 1 cm grid behind the flask. In total, the movement patterns of 284 zebrafish were recorded and measured: 41 of 4 dpf WT, 130 of 4 dpf KD, 32 of 5 dpf WT, and 81 of 5 dpf KD. Sample sizes of at least 30 per group were chosen a priori because normality of distributions could not be assumed.

The movements of all of the larvae were analyzed using the MtrackJ plugin (Meijering, Dzyubachyk & Smal, 2012) within the ImageJ program (Schneider, Rasband & Eliceiri, 2012). Tracking and recording of fish movements and analysis of movement data were assigned to two separate researchers to avoid biasing the analysis. Distances traveled (cm per 1 min, for Fig. 3A) and time spent in the upper half of the tank (seconds, out of 60 s, for Fig. 3B) were calculated for each fish, compiled by group, and presented as boxplots using the Matplotlib (Hunter, 2007) Python library. Statistical testing of similarity between each group, using the Kolmogorov–Smirnov two sample test, was performed using the Stats module of the SciPy Python library (van der Walt, Colbert & Varoquaux, 2011). The two-sample Kolmogorov–Smirnov test was chosen because it makes no assumption about the distribution of data.

Figure 3 Zebrafish wild-type and ca6 knockdown movement analysis.

Boxplots from video analysis of 1 min of swimming of zebrafish larvae. KD, knockdown; WT, wild-type. Same data used for both (A and B). Statistics of both analyses are given in Table 3. (A) Total distances traveled. (B) Time spent in the upper half of the tank (seconds, out of 60 s).

Results

Non-mammalian CA VI contains an additional PTX domain

We retrieved 78 CA VI protein sequences from 75 non-mammalian species in NCBI GenPept, all of which have the C-terminal PTX domain. The PTX domain in CA VI is less conserved than the CA domain. The multiple sequence alignment of the 78 CA VI sequences (Fig. S1) shows that there are 83 perfectly conserved amino acids within the catalytic domain (within MSA columns 30–288), whereas only 19 amino acids in the PTX domain are perfectly conserved (within MSA columns 355–566). The region between the CA and PTX domains consists of a moderately conserved and gapless region (MSA columns 300–320) flanked by two highly variable regions of flexible length (MSA columns 292–297 and 326–344). The presence of CA and PTX domains in non-mammalian CA VI sequences has also been documented in the Pfam database since many years (Finn et al., 2016), for example in http://pfam.xfam.org/protein/E9QB97_DANRE.

Figure 1 presents the phylogenetic tree of CAs VI, IX, XII, and XIV, clearly showing that the longer, non-mammalian isoforms (with a PTX domain) are orthologs of mammalian CA VI. The pairwise arrangement of VI/IX vs. XII/XIV is the same as in previous phylogenetic work (Hewett-Emmett, 2000), suggesting that these four CA isozymes descend from one common ancestor. Figure 2 shows a phylogenetic tree of all human PTXs and selected CA-linked PTX domains, which indicates that the novel PTX domains would be most closely related to the short PTXs, CRP and APCS or SAP.

Platypus (Ornithorhynchus anatinus) is probably an exception in the pattern of mammals not having a PTX domain associated with CA VI

A genomic fragment not assigned to any chromosome (Contig22468 in assembly WUGSC 5.0.1/ornAna1) contains an exon which codes for a PTX domain unlike any that we find in other mammalian species, and most similar to CA VI-linked PTX domains in non-mammalian species. The phylogenetic tree in Fig. 2 demonstrates that this platypus PTX sequence is orthologous with the PTX sequences associated with CA VI in non-mammalian species. What is more, a BLASTN search of Contig22468 against the platypus genome showed that it partially matches a region in chromosome 5 right after the CA6 locus. More specifically, the first 703 bp of Contig22468 match the last 703 bases (99.86% identity, a single mismatch) of Contig3933.5.

The adjacent location of Contig3933.5 to Contig3933.4, the fragment containing the exons coding for the CA6 ortholog (ENSOANG00000013215), would put the exon coding for the PTX domain in the correct location and orientation to be part of the platypus CA6 gene if Contig22468 were placed in this position. Therefore, we tentatively label this PTX domain as “CA-linked” and suggest that Contig22468 would be more correctly mapped starting from Chr5:18954728 in platypus genome assembly OANA5. With this evidence, we also suggest that CA VI in platypus contains a PTX domain, and consequently, that the loss of PTX domain occurred after the separation of monotreme and therian lineages in mammals.

One further phylogenetic tree was made based on CA domain sequences, showing that phylogeny of CA VI follows the expected vertebrate phylogeny, with platypus placed outside of marsupials and placental mammals (Fig. S2).

Exon lengths suggest that the region after the CA domain in CA VI descends from the transmembrane helix of the ancestral form

Mammalian CA VI proteins contain an additional C-terminal region of at least 25 residues, which is dissimilar to anything in other vertebrate CA isoforms and of unknown structure. Non-mammalian CA VI contains a sequence homologous to this extension as a spacer region between the CA and PTX domains. In order to investigate the most likely origin of the spacer region, we compared the exon lengths in CA6 and the most closely related CA genes (CA9, CA12, and CA14) and short PTXs. The length of the exon coding for the spacer between CA and PTX domains in zebrafish ca6 is 84 bp, and the coding sequence of the homologous exon in human CA6 is 83 bp. The exons coding for the region containing the transmembrane (TM) helices (penultimate exons) in CA9, CA12, and CA14 are 82, 85, and 85 bp in length, respectively. Assuming a novel juxtaposition of exons between genes coding for the ancestral TM form of CA6 and a short PTX, the final exon of CA6 and the first exon of the PTX gene are less likely to have been retained. Because they contain non-coding UTR sequences and lack splice donor and acceptor sites, they would be unlikely to be spliced correctly as continuous, protein-coding sequence. Taken together, this suggests that only the exon coding for the cytoplasmic domain of ancestral CA VI was lost and replaced by the single exon coding for the PTX domain. This also implies that the last exon in mammalian CA6 and the penultimate exon of non-mammalian CA6, predicted to code for an APH (see below), and the penultimate exons of CA9, CA12, and CA14, coding for the TM helix, are highly likely to share a common ancestry.

The region after the CA domain is predicted to contain an APH

The pattern of hydrophobic residues repeating approximately every fourth residue is obvious in the alignment of the region following the CA domain (final domain in mammalian CA VI, or the segment between CA and PTX domains in non-mammalian CA VI), as seen in Fig. 4C and in the larger alignment of Fig. S1. The helical wheel visualizations of Figs. 4A and 4B indicate that when folded as an alpha helix, this region of human and zebrafish CA VI, respectively, would be an APH, with one side lined with mainly hydrophobic residues (in blue and lilac). Furthermore, this region (292–312) in zebrafish CA VI is also predicted to have a high potential to form a coiled-coil structure by the COILS algorithm (Lupas, Van Dyke & Stock, 1991) in InterProScan at http://www.ebi.ac.uk/interpro/sequence-search (Jones et al., 2014). The APH region is a unique feature of CA VI, present in both non-mammalian and mammalian sequences.

Figure 4 Amphipathic helix analysis in CA VI.

(A) Helical wheel diagram of human CA VI (287–303). (B) Helical wheel diagram of zebrafish CA VI (293–310). Multiple sequence alignment of the spacer region of CA VI from indicated species. (C) CA indicates the end of the catalytic CA domain; APH is the suggested amphipathic helix, which is analyzed in (A) and (B); and PTX indicates the approximate start of the pentraxin domain (not applicable to Homo sapiens CA VI).

Duplication of an adjacent glucose transporter gene is associated with the loss of PTX from CA VI

The genes next to CA6 provide a clue for a possible cause of losing the PTX-encoding exon in mammalian CA6. We have observed 17 non-mammalian genomes with a chromosomal arrangement of CA6, then one glucose transporter gene (SLC2A5/SLC2A7), followed by the gene GPR157, whereas most mammalian genomes present the gene order CA6, SLC2A7, SLC2A5, and GPR157. The reconstructed syntenic block for therian mammals in the region after CA6 in Genomicus (http://www.genomicus.biologie.ens.fr/genomicus-86.01) (Muffato et al., 2010) also shows the duplicated glucose transporter, whereas those for ancestral tetrapods and bony fish lineages only have a single SLC2A5/SLC2A7 ortholog. We were not able to find any single genome containing a PTX-coding exon with CA6 and both SLC2A5 and SLC2A7. Hence, the available genomic evidence suggests that the loss of the PTX-domain-coding exon and the duplication of the adjacent glucose transporter gene may have occurred simultaneously, close to the divergence time of the mammalian lineage. The rearrangements during the gene duplication would also provide a plausible mechanism for the exon loss.

Sequencing of zebrafish ca6 cDNA confirms a 530-residue product

We produced a PCR-amplified cDNA of zebrafish ca6 for recombinant protein production. The resulting sequence had five synonymous substitutions compared to Ensembl ENSDART00000132733 (Fig. S4) and three unresolved bases leading to one unknown amino acid residue. Except for the unknown residue, the translation is identical to the predicted 530-residue protein (Ensembl ENSDARP00000119189 or UniProt E9QB97, Fig. S5). The cDNA sequence has been submitted to ENA database (http://www.ebi.ac.uk/ena) as LT724251 and its translation to UniProt as A0A1R4AHH7. The other predicted Ensembl transcript (ENSDART00000079007) codes for a protein of 538 residues, in which an additional 24 bp exon creates an insertion before the PTX domain.

Sequence alignment predicts three disulfides in zebrafish CA VI

Cysteine pairs 44/226 in (CA domain, MSA columns 51/234 in Fig. S1), 352/408, and 487/518 (PTX domain, columns 390/453 and columns 532/564, respectively in Fig. S1) are expected to form disulfides by sequence conservation in the multiple sequence alignment. All three disulfides are also structurally verified. The one in CA domain is seen in all structures of extracellular CAs, e.g., human CA VI in PDB 3FE4 (Pilka et al., 2012), and the disulfide 352/408 in the PTX domain is homologous to the one in short PTXs, e.g., human CRP in PDB 3PVN (Guillon et al., 2014). The third disulfide, 487/518, is also supported by proximity in our molecular model (Fig. 5A). There is one further unpaired Cys290, in the region between the CA and PTX domains (and missing from the model), which is also conserved in 76 of 78 non-mammalian sequences (Fig. S1).

Figure 5 Molecular models of zebrafish CA VI–PTX.

(A) One protomer shown in two orientations, CA domain at the top, PTX at the bottom. Potential glycosylation site Asn residues are shown as spheres, active-site histidines and assumed disulfide cysteines as sticks. (B) Front view of the pentamer model. In (B–D), individual protomers are shown in different colors. Asn residues in glycosylation sites and active-site histidines are shown in spheres, cysteines not highlighted. (C) Back view of the pentamer model. (D) Side view of the pentamer model, seen from the top of (C), with back view downwards.

3D model of zebrafish CA VI–PTX is compatible with predicted APH and disulfides

We made a homology-based model of the CA and PTX domains and combined it with an alpha helical model of the predicted APH region, using protein–protein docking to create the nearly full model. Figure 5A shows the model, in two orientations, with the CA domain at the top and the PTX domain at the bottom. The APH (pink) fills a non-polar cavity on the surface of the CA domain. The precise orientation of the PTX domain is impossible to predict with certainty, but the current model shows it leaning against the CA domain and APH. The most highly variable regions, for which no template was available, were not modeled (residues 281–292 and 311–317), indicated by yellow dotted lines (Fig. 5A). In addition, the N-terminus of the model is incomplete, missing residues 20–31, which are not visible in the template 3FE4.

The zinc-binding histidines in the active site of the CA domain are shown as yellow sticks in Fig. 5A (zinc not shown), with the active-site cavity opening upwards. Disulfide-forming cysteines are presented as orange stick models. The disulfide in the CA domain and the one in the beta sheet of the PTX domain (lowest in Fig. 4A) are also present in the templates. The information of the predicted third disulfide on the surface of the PTX domain was not used when building the model, but the cysteines ended in close proximity so that the disulfide could be constructed by minor refinement of the model. This disulfide would lock the C-terminus of the PTX domain on the surface of the domain. The presumably unpaired Cys290 is part of an unmodeled region.

Based on the pentamerization tendency of mammalian PTX domains, we constructed an additional pentameric model of zebrafish CA VI (Figs. 5B–5D) by superimposing the PTX domains of five copies of the monomer model on the pentameric structure of SAP (PDB 4AVS). Individual monomers are presented in different surface colors. There are no serious steric clashes in the model, and the domain axes align to make a flat pentamer complex (Fig. 5D), even if no pentamer constraints were applied for the monomer model. Furthermore, adjacent monomers form an additional protein–protein interface between the sides of their PTX and CA domains. The general shape of the modeled pentamer is a flat, roughly planar five-pointed star, thickness 4–5 nm and an approximate diameter 15 nm. The active site of CA faces outward in the pentamer so that the zinc-binding histidines (yellow spheres in panels B–D) are exposed in the active-site cavity, as seen in the center of panel D.

The four potentially N-glycosylated Asn residues (in the motif Asn-X-Ser/Thr) are all on the surface of the monomer, shown as spheres in Fig. 4A. In contrast, the pentamer model only shows three of them on the surface of the pentamer. Asn210, shown in cyan, is buried between the monomers, conforming well with the observed non-glycosylated status for this Asn residue. The coloring of the potential glycosylation sites in Figs. 4A–4D reflects their observed glycosylation status (presented below under mass spectrometry).

Recombinant CA VI–PTX shows a high catalytic activity

Zebrafish CA VI–PTX was produced in insect cells with high yield. The purified protein showed a single band close to the expected size in SDS-PAGE (Fig. 6, measured MW 58.6 kDa, theoretical 58.107 kDa without glycans, signal peptide excluded). Carbonate dehydratase activity was analyzed kinetically in the presence or absence of acetazolamide. The kinetic parameters of CA VI–PTX (kcat and kcat/Km) were then compared with those of the thoroughly investigated CAs, namely the cytosolic and ubiquitous human isozymes α-CA I (hCA I) and II (hCA II). The CA VI–PTX possesses considerable carbonate dehydratase activity as shown in Table 2. A kcat of 8.9 × 105 s−1 and a kcat/Km of 1.3 × 108 M−1 × s−1 show that the enzymatic activity of CA VI–PTX is almost in the same range with the very highly active human CA II. Data also show that CA VI–PTX was efficiently inhibited, with an inhibition constant of 5 nM, by the clinically-used sulfonamide, acetazolamide (5-acetamido-1,3,4-thiadiazole-2-sulfonamide).

Figure 6 SDS-PAGE of recombinantly produced zebrafish CA VI–PTX.

Left: purified recombinant zebrafish CA VI–PTX, molecular mass calculated from mobility 58.6 kDa. Right: molecular weight standards.

Table 2 Kinetic parameters for CO2 hydration reaction catalyzed by selected α-CA isozymes.

Enzyme	kcat (s−1)	Km (mM)	kcat/Km (M−1 s−1)	KI(AAZ) (nM)	
hCA Ia	2.0 × 105	4.0	5.0 × 107	250	
hCA IIa	1.4 × 106	9.3	1.5 × 108	12	
Pentraxin-CA VIb	8.9 × 105	6.5	1.3 × 108	5	
Notes:

AAZ, acetazolamide, 5-acetamido-1,3,4-thiadiazole-2-sulfonamide.

a Human recombinant isozymes, stopped flow CO2 hydratase assay method (pH 7.5) (Nishimori et al., 2007).

b Zebrafish recombinant enzyme, stopped flow CO2 hydratase assay method (pH 7.5), this work.

Light scattering analysis by LC–SLS–DLS confirms multimeric structure

The molecular size of native recombinantly produced zebrafish CA VI–PTX was estimated by SLS and DLS analysis after liquid chromatography. Gel filtration analysis indicated main peak eluting at 1.52 ml retention volume according to A280 (Fig. 7, black curve). This was associated with SLS intensity peak with identical shape. Analysis of the scattering intensity (SLS) results in a MW estimate of 280 ± 11 kDa for the peak, and the estimate was homogeneous throughout the elution peak (Fig. 7, near-horizontal line across the peak in dark gray). In addition, DLS data was collected for the eluted peak indicating particle size of 7.69 ± 0.29 nm, as Rh (hydrodynamic radius), which is consistent with the determined MW. The MW estimate based on the retention volume in gel filtration is slightly smaller (214 ± 10 kDa), possibly due to off-globular shape of the molecule. The small peak eluting before the main peak (∼1.1 ml retention volume) indicated the presence of aggregated protein, resulting in high scattering intensity. According to A280, this is less than 5% of the protein sample. Altogether, the light scattering analysis combined with gel filtration indicates oligomeric assembly for the protein, a pentameric form being the most probable oligomeric state.

Figure 7 Assessment of the oligomeric size of zebrafish CA VI.

Gel permeation chromatography was used to study the characteristics of recombinantly produced zebrafish CA VI. The left Y-axis shows the UV absorption intensity (280 nm wavelength) and light scattering (LS) intensity. UV intensity was used for the determination of the protein concentration. Molecular weight (MW) was calculated using LS intensity and shown on the right Y-axis. Hydrodynamic radius (Rh) was calculated from the dynamic light scattering signal, and is also shown on the right Y-axis. In addition, the oligomeric size of zebrafish CA VI was evaluated based on the penetration time using molecular weight marker proteins as a standard.

Mass spectrometry confirms post-translational modifications

All attempts to characterize the intact CA VI–PTX with ESI FT-ICR mass spectrometry failed, despite the extensive sample desalting/purification prior to the measurements. This may be due to a slight protein precipitation observed during the sample preparation. Therefore, in-solution trypsin digestion was selected as the main route for structural characterization of CA VI–PTX. The digestion was performed in non-reducing conditions to preserve disulfide bonds in the structure. The digestion resulted in 97% sequence coverage with 64 specific tryptic peptides identified (Figs. 8 and 9, and fuller details in Fig. S6 and Table S1).

Figure 8 High-resolution mass spectrum of the tryptic digest of CAVI-PTX.

The mass spectrum was measured by direct infusion on a 12-T Fourier transform ion cyclotron resonance instrument using positive-ion electrospray ionization. Monoisotopic m/z values and charge states obtained through peak deconvolution are indicated for the most abundant peaks.

Figure 9 A tryptic peptide map of selected peptides of zebrafish CA VI–PTX.

The identified tryptic peptides are indicated with blue boxes showing the start and the end residues. The confirmed disulfide bonds are indicated with red lines with the corresponding peptides indicated with red boxes. The four potential N-glycosylation sites are indicated with a distinct background color (blue: unoccupied N-glycosylation site; and green: occupied N-glycosylation site). The three observed glycopeptides are marked with purple boxes. The start of the PTX domain (KQP…) has been indicated with a black arrow. This figure shows a minimum amount of peptides for maximal coverage, whereas all identified peptides are shown in Fig. S6.

The peptide map in Fig. 9 shows that the tryptic peptides were found within both protein domains, although somewhat larger peptides (up to ∼14 kDa) were found within the PTX domain. Out of all identified peptides, 12 contained disulfide bonds (either intra- or interpeptide). These peptides confirmed the putative disulfide bonds, Cys 44/226 in the CA domain, and Cys 352/408 and Cys 487/518 in the PTX domain. Cys290 in the spacer region is most likely free but the corresponding tryptic peptide (LSKGGMCR) was not observed to confirm this. These disulfide bonds are fully consistent with the 3D structural model of CA VI–PTX.

Carbonic anhydrase VI–pentraxin contains four putative N-glycosylation sites (Asn210, Asn258, Asn339 and Asn394), having a canonical NxS/T consensus sequence (marked in Fig. 9). Among the identified tryptic peptides, 12 glycopeptides were found. On the basis of these peptides, CA VI–PTX carries two glycans, a core-fucosylated oligomannose type glycan GlcNAc2(Fuc)Man3 at Asn258 and an oligomannose type glycan GlcNAc2Man3 at Asn339, located in the CA domain and PTX domain, respectively. These glycosylation sites and glycan structures were further verified by CID-MS/MS experiments of the representing glycopeptides [248–266] (3416.5084 Da) and [331–347] (2819.3059 Da) (Fig. 10). As no other glycan variants were observed among the peptides, it seems that the glycosylation in CA VI–PTX (produced in insect cells) is rather homogenous. These results are consistent with accessibility of the sites predicted by our model. Interestingly, the glycosylation site at Asn258 is conserved in 77 out of 78 non-mammalian CA VI sequences in the sequence alignment Fig. S1 (columns 266–268). The tryptic peptide [191–216] (2988.4744 Da) was only observed in a free form, indicating that Asn210 is non-glycosylated in the CA domain. Similarly, the peptides spanning the Asn394 residue were all observed without any glycans attached (Fig. S6), suggesting that this site is non-glycosylated in the PTX domain.

Figure 10 Characterization of zebrafish CA VI–PTX glycopeptides by tandem mass spectrometry.

The precursor ions of the two observed glycopeptides, with monoisotopic masses of 3416.5084 Da and 2819.3059 Da (residues 248–266 and 331–347, respectively), were mass-selected in a quadrupole for collision-induced dissociation tandem mass spectrometry. The fragmentation patterns are consistent with the presence of the standard N-glycosylation core pentasaccharide with fucosylation in the innermost N-acetylglucosamine residue in the glycopeptide 248–266 (A) and a similar non-fucosylated pentasaccharide in the glycopeptide 331–347 (B).

Immunohistochemistry shows cell surface localization of CA VI–PTX in various tissues

Recombinant zebrafish CA VI–PTX protein was used to raise a rabbit polyclonal antiserum, which worked well in immunofluorescence studies. Figure 11 shows positive staining in the skin, heart, gills, and swim bladder. The strongest signal is seen on cell surfaces, while the intracellular staining was detectable but weaker.

Figure 11 Immunohistochemistry of CA VI–PTX in adult zebrafish tissues.

Tissue sections stained by anti-zebrafish CA VI–PTX (green) and nuclear staining by DAPI (blue). Gills (A), heart (B), skin (C), swim bladder (D). The strongest signal (A–D) is present on the cell surfaces, even though the cell interior gives some background staining. Scale bars 100 μm.

To get further insights into ca6 expression in zebrafish, we also studied the expression pattern in different tissues of adult zebrafish by qRT-PCR. As shown in Table 3, relative expression of ca6 mRNA was found to be prominent in the fins/tail, and brain. Low levels of expression were observed in the gills, kidney, teeth, skin, and spleen. A very faint signal was detected in the swim bladder, intestine, pancreas, liver, eggs, and heart.

Table 3 Relative expression ratios of ca6 mRNA in adult zebrafish tissues.

Tissue	Relative expression	
Fin/tail	214.82	
Teeth	6.81	
Spleen	2.37	
Kidney	19.44	
Brain	293.46	
Swim bladder	0.61	
Heart	0.01	
Gills	68.70	
Skin	4.32	
Intestine	0.41	
Pancreas	0.27	
Eggs	0.01	
Liver	0.02	

Zebrafish cannot swim properly in the ca6 knockdown model

Gene-specific antisense MOs have been widely used to inhibit gene expression in zebrafish larvae (Eisen & Smith, 2008). We designed two different MOs, one for translational blocking of ca6 mRNA and the other for blocking intron splicing before exon 9. Both MOs were used to repeat all knockdown experiments with highly similar results, suggesting equally efficient knockdown in both kinds of ca6 morphants. We did not see any morphological differences between uninjected and RC MO-injected embryos/larvae over the period of five days of development. The ca6 morphant zebrafish embryos between 1 and 3 dpf were also devoid of any notable morphological changes, but interestingly, at the end of 4 dpf we consistently observed an underdeveloped or deflated swim bladder in ca6 morphant larvae (Fig. 12).

Figure 12 Comparison between morpholino-injected and wild-type zebrafish larvae.

The morphant larvae (A) showed consistently a deflated swim bladder at 4 dpf (arrow), which returned to normal morphology at 5 dpf. Wild-type larvae of the same ages are shown for comparison (B). Scale bars 1 mm.

The quantitative expression analysis of ca6 mRNA was done in wild-type and the ca6 morphant zebrafish at different stages of development. As seen in Fig. 13, the mRNA expression in wild-type embryos was highest at 24 hpf, with slightly lower values later. The levels of ca6 mRNA were consistently higher in the morphant embryos compared to the wild-type, possibly because of compensatory upregulation of the gene caused by the absence of CA VI protein. The peak expression of ca6 was at 48 hpf in the ca6 morphant embryos.

Figure 13 Developmental expression pattern of ca6 in 1–5 dpf larvae.

The expression levels of the ca6 gene was studied using qRT-PCR from the total mRNA isolated from 1 to 5 dpf of ca6 morphant and wild-type larvae. The results of ca6 gene expression were normalized using gapdh as internal control.

In order to measure swimming activity of morphant vs. wild-type, we calculated total distances traveled for individual larvae, and they are presented as boxplots in Fig. 3. Two-sample Kolmogorov–Smirnov statistical analyses were performed between relevant group pairs to determine if they could have been drawn from the same distribution. Day 4 knockdown larvae swam less (median 0.00 cm) than day 4 wild-type larvae (median 13.80 cm, p-value 4.28 × 10−19), and similarly day 5 knockdown larvae swam less (median 4.75 cm) than day 5 wild-type larvae (median 10.22 cm, p-value 1.16 × 107). Full details of the swimming data are shown in Table 4. Taken together with the clearly observed swim bladder deficiency in 4 dpf larvae (Fig. 12) and the presence of CA VI in adult zebrafish swim bladder, we suggest that CA VI is required either for swim bladder development or swim bladder function. When CA VI expression is mainly restored in 5 dpf larvae, the swimming pattern also returns to almost normal.

Table 4 Statistics of swimming pattern analysis of ca6 morphant and wild-type zebrafish.

Aa	Median	Mean	SD	Range	p-Value	
Day 4	
KD	0.00	1.87	6.59	0.00–49.03	4.28 × 10−19 b	
WT	13.80	13.12	6.01	0.00–25.45	1.90 × 10−3 c	
Day 5	
KD	4.75	4.89	4.26	0.00–5.19	1.16 × 10−7 b	
WT	10.22	10.38	3.18	1.83–17.27		
Bd	Median	Mean	SD	Range	p-Value	
Day 4	
KD	0	9.13	20.46	0.00–60.00	8.68 × 10−11 b	
WT	31	29.59	22.44	0.00–60.00	4.98 × 10−3 c	
Day 5	
KD	24.51	26.96	26.23	0.00–60.00	2.98 × 10−4 b	
WT	51.47	45.29	16.42	0.00–60.00		
Notes:

KD, knockdown; WT, wild-type.

a Swimming distances (cm).

b KD compared to WT.

c Day 4 WT compared to day 5 WT.

d Time spent in upper half of the flask.

Discussion

This study consists of the characterization of a novel type of a CA, CA VI containing a PTX domain, by means of sequence analyses, phylogenetics, molecular modeling, experiments on a recombinantly produced protein, knockdown of the ca6 gene in zebrafish embryos, and expression studies by immunohistochemistry and qRT-PCR. The bioinformatic and experimental analyses build a coherent picture of the structure of this novel domain combination, and the evolutionary analysis shows a history of domain gains and losses. Based on our previous work and the findings in this study, we propose that CA VI–PTX in zebrafish is needed for filling the swim bladder, and possibly in a novel type of membrane anchoring and immune function.

The PTX domain found associated with non-mammalian CA VI is a novel member of the PTX family. We have shown it to be most closely related with the short PTXs, CRP and SAP (Fig. 2). The association of a CA domain with a PTX domain is new in both the PTX and CA families. SAP and CRP are more closely similar to each other than either is to the CA-associated PTX domain. This could indicate that the CA-associated PTX domain had diverged from a common ancestor before the duplication that created SAP and CRP, but we cannot take this for granted, because adaptation to create a viable domain interface may have accelerated the rate of change in the CA-associated PTX domain.

The phylogenetic tree in Fig. 1 shows that the TM CAs IX, XII, and XIV and secretory CA VI share a common ancestor. We propose that the quartet has arisen in the two whole-genome duplications in early vertebrates. Figure 14 presents a plausible sequence of events that could have led to present-day domain structures in CA VI. Briefly, we assume that the exon coding for the cytoplasmic domain in ancestral CA VI was replaced by an exon coding for a PTX domain (probably by a duplication or a move of an exon coding for a short PTX in early vertebrates), and the TM helix transformed into an APH (Figs. 4 and 5). Later, presumably in the therian mammal lineage, the PTX domain was lost, leaving the APH in the C-terminus of CA VI. These hypotheses are supported by the following observations: (1) Comparison of exon lengths suggests the TM-helix-coding exon as the most likely ancestor of the exon coding the spacer region after the CA domain in CA VI; (2) the losses of the CP domain in early CA VI and of PTX domain in mammalian lineage are more parsimonious assumptions than their acquisition in multiple lineages; (3) the duplication of the glucose transporter genes SLC2A5 and SLC2A7, as seen in therian mammals, is evidence of rearrangements in the region adjacent to the PTX-domain-coding exon of the CA6 locus, which we assume to have led to the loss of the PTX domain in mammalian CA VI; and (4) the PTX domain is consistently present in non-mammalian CA VI and missing from mammalian CA VI (most likely excepting platypus).

Figure 14 Hypothesis of evolution of the domain composition in CA VI and the transmembrane CA isoforms.

CA, catalytic CA domain; TMH, transmembrane helix; APH, amphipathic helix; PTX, pentraxin domain; PG, proteoglycan domain. Image credit: Original digital art by Jukka Lehtiniemi.

Considering the monomer MW of 58.1 kDa (plus glycosylation), the LC–SLS–DLS results clearly confirm that zebrafish CA VI is an oligomer. The MW estimated by LC–SLS (280 ± 11 kDa) is slightly less than MW calculated from sequence (290.5 kDa for pentamer, plus glycosylation). Based on the gel filtration retention volume and protein standards, the MW is estimated to be slightly smaller (214 ± 10 kDa), but this result may be affected by column interactions and deviation from the globular shape. Furthermore, the hydrodynamic radius calculated from light scattering (7.69 ± 0.29 nm; diameter 15.38 ± 0.58 nm) suggests a particle size in the range of 364–434 kDa for globular particle. In this context, it has to be noted that diffusion of the particle is highly dependent on the molecular shape and DLS-based estimate may also be slightly affected by irregular shape. Taken together, the light scattering results are more compatible with a pentamer than tetramer or hexamer models. The 3D model of CA VI–PTX as a pentamer (Figs. 5B–5D) predicts a shape of a flat, roughly planar five-pointed star, thickness 4–5 nm and approximate diameter 15 nm, i.e., clearly off-globular, which would explain the minor conflicts between observations. What is more, the pentamer model is also supported by known pentamerization of related PTXs (CRP and SAP). However, we need to stress that the relative orientations of the CA and PTX domains in our models are only tentative.

Mass spectrometry confirms that the N-terminus of the mature CA VI–PTX coincides with the predicted signal peptide cleavage site between residues 19 and 20. Glycopeptides with typical N-linked glycans are observed associated with Asn258 and Asn339, whereas the peptides containing Asn210 or Asn394 are only seen in non-glycosylated form (Fig. 9). Consistent with these observations of N-glycosylation, our 3D model of pentameric CA VI–PTX (Figs. 5B–5D) shows that Asn258 and Asn339 are well exposed, whereas Asn210 is fully buried in the protomer/domain interface, and Asn394 would be somewhat hindered at the protomer interface.

We discovered a minor but surprising outcome in the knockdown model regarding the poor floating ability, most likely caused by a deflated swim bladder, both of which we observed consistently in 4 dpf knockdown larvae. The statistically significant lower swimming distances and stationary positioning at the bottom of 4 dpf knockdown larvae, vs. those of 4 dpf wild-type larvae or 5 dpf knockdown larvae, imply that the knockdown larvae gain normal swimming function as the knockdown action of the injected MOs is relieved (Fig. 12). CA VI–PTX function within the swim bladder is further supported by immunohistochemistry and qRT-PCR, showing expression of both CA VI–PTX protein and mRNA in the swim bladder specimens. However, at the current point we cannot distinguish whether the swim bladder dysfunction observed in 4 dpf larvae is due to delayed development or the need of CA VI–PTX in swim bladder inflation.

C-reactive protein and serum amyloid P are known to bind carbohydrates, i.e., they are lectins (Hind et al., 1984; Kottgen et al., 1992). The calcium-binding residues in the sugar binding site are partially conserved between these two lectins and the CA-associated PTX domain. In addition, PTXs are a subfamily of the Concanavalin A-like lectin/glucanase family, which contains numerous other lectins (leguminous plant lectins, animal galectins, etc.) and other proteins interacting with carbohydrates (http://www.ebi.ac.uk/interpro/entry/IPR013320). In our immunohistochemistry results the CA VI–PTX protein shows mostly a strong cell-surface staining pattern (Fig. 11), even if the protein is predicted to be a secreted, soluble protein. We assume that the PTX domain in CA VI would also be a lectin and anchor the protein on the cell surface via sugars in glycoconjugates. Binding to plasma membrane glycoconjugates would also explain why the loss of the TMH was tolerated, i.e., TM helix anchoring was replaced by lectin anchoring. If sugar binding by CA VI–PTX can be proved experimentally, non-mammalian CA VI would represent the first case of an enzyme which is attached on the cell surface by lectin binding.

Lectins and other PRMs are an important part of the innate immune system in fishes, which is more diverse than that of mammals (Vasta et al., 2011; Sunyer, Zarkadis & Lambris, 1998). Although teleost fish lack lymph nodes and bone marrow, the anterior part of the fish kidney is considered a functional ortholog of mammalian bone marrow. Thus, it represents the main hematopoietic lymphoid tissue of teleosts, and is thought to be an immunologically responsive organ (Zapata & Amemiya, 2000). The role of maintenance of mucosal homeostasis is served in teleosts by the gut, skin, and gills, which all contain mucosa-associated lymphoid tissue (Salinas, Zhang & Sunyer, 2011). These are among the tissues where zebrafish CA VI–PTX has its highest expression, and therefore we assume that this protein is a part of the innate immune system.

Interestingly, we have shown that mouse CA VI is also highly expressed in the gut, specifically in the immunologically active Peyer’s patches (Pan et al., 2011). In another study, we demonstrated that there is a likely role for Car6 in immune stimulated lung tissues (Patrikainen et al., 2016) and murine Car6 is likely involved in mucosa maintenance in both airways and gut (Leinonen et al., 2004; Parkkila et al., 1997). We formed a preliminary hypothesis that mouse CA VI is involved in immunological functions, which has been confirmed recently (Xu et al., 2017), by showing that CA VI isoform B promotes interleukin-12 expression. However, a gene regulatory function is unlikely for zebrafish CA VI, with the estimated diameter of 15 nm for the pentamer making it too large to enter the nuclear pores. The locations of high ca6/CA6 expression in fish and in mammals are similar in that they allow delivery of CA VI on the physical barrier against external environment (gut, skin, and gills in zebrafish; skin, saliva, milk, and lungs in human/mouse), consistent with a function associated with primary immune defense. Summing up, we suggest that both mammalian and fish CA VI are components of the innate immune system, with or without a PTX domain.

Given the dynamic nature of genomes, with transposition and translocation events constantly shuffling exons, it is hard to see the choreography of domain moves in CA VI during vertebrate evolution as anything more than chance events. However, in order to remain stably in a genome, the changes must be at least tolerated, or possibly provide some advantage to their carrier. We see the addition of the PTX domain in early jawed vertebrates as a tolerated change, in which membrane attachment through a TM helix was replaced by lectin anchoring. As we have suggested, the new domain context may have led to the CA domain of CA VI adopting functionality within the innate immune system. Then later, when the PTX domain was lost, presumably through the local segmental duplication leading to a duo of glucose transporters (SLC2A5 and SLC2A7), the addition of another glucose transporter may have been more of an advantage than the loss of pentamerization and membrane anchoring in CA VI, and thus this chromosomal arrangement became fixed in early therian mammals. The loss of the PTX domain may also have opened the way for using the APH in forming dimers. We have a preliminary result of human CA VI being a mixture of monomer and dimer forms in solution (A. Yrjänäinen, 2017, unpublished data), in which we speculate dimerization to be mediated by the amphipathic helices being able to join in a coiled-coil fashion when unhindered by a further C-terminal domain.

This study has given us many ideas for future research. We plan to take a closer look at the complex evolution of non-mammalian PTXs, which might shed more light on the origin of the CA VI-linked PTX domain and on structure-related constraints on its surface. We have also started work on comparisons of per-residue conservation patterns of the CA domain in mammalian vs. non-mammalian CA VI. Testing the sugar-binding ability of CA VI–PTX will be the obvious way to explore the lectin hypothesis.

Supplemental Information

Supplemental Information 1 Multiple protein sequence alignment of Fig. S1 in Fasta format.

Raw data.

Click here for additional data file.

Supplemental Information 2 Multiple protein sequence alignment for phylogenetic tree of Fig. 1 in Clustal format.

Raw data.

Click here for additional data file.

Supplemental Information 3 Multiple codon sequence alignment for phylogenetic tree of Fig. 1 in Clustal format.

Raw data.

Click here for additional data file.

Supplemental Information 4 Multiple protein sequence alignment for phylogenetic tree of Fig. S2 in Clustal format.

Raw data.

Click here for additional data file.

Supplemental Information 5 Multiple codon sequence alignment for phylogenetic tree of Fig. S2 in Clustal format.

Raw data.

Click here for additional data file.

Supplemental Information 6 Multiple sequence alignment for phylogenetic tree of Fig. 2 in Fasta format.

Raw data.

Click here for additional data file.

Supplemental Information 7 Multiple sequence alignment of 78 non-mammalian CA VI sequences.

Four of the sequences were edited as specified in Table 4.

Click here for additional data file.

Supplemental Information 8 Bayesian phylogenetic tree of CA domains of CA VI of indicated species.

Analysis of protein alignment guided DNA alignments as detailed in Materials and methods. Sidebar indicates species in which the presence of a PTX domain in CA VI is observed (non-mammals) or assumed (O. anatus).

Click here for additional data file.

Supplemental Information 9 Dot blot analysis of rabbit anti-zebrafish CA VI-PTX serum.

500 ng of purified recombinant zebrafish CA VI-PTX antigen was attached as spots on nitrocellulose, incubated with antisera from various stages of the immunization process, and stained with a peroxidase-coupled secondary antibody. A) Pre-immune serum, no recognition of the CA VI-PTX protein; B) Bleed 1 (day 41); C) Bleed 2 (day 62); and D) Bleed 3 (day 83). In B and C the antiserum strongly recognizes the recombinant protein, whereas the signal is clearly diminished when bleed 3 (D) was tested.

Click here for additional data file.

Supplemental Information 10 Alignment of the sequenced zebrafish CA VI-PTX cDNA with the sequence from Ensembl database.

ca6zf = sequence of the PCR amplification product of this study [submitted to ENA database (http://www.ebi.ac.uk/ena) as LT724251], CA6 = coding sequence of transcript ENSDART00000132733 from Ensembl. Translated sequences are shown aligned in Fig. S5.

Click here for additional data file.

Supplemental Information 11 Translation of the sequenced zebrafish CA VI-PTX cDNA.

Comparison to a reference sequence. ca6zf = protein translation of the PCR amplification product of this study (UniProt A0A1R4AHH7), CA6 = reference protein sequence (UniProt E9QB97). Underlined N-terminal sequences: Signal peptides as reported in UniProt and as confirmed by mass spectrometry in this study.

Click here for additional data file.

Supplemental Information 12 Tryptic peptide map of zebrafish CA VI-PTX with all identified peptides.

Click here for additional data file.

Supplemental Information 13 List of peptides from zebrafish CA VI-PTX observed by mass spectrometry.

Click here for additional data file.

Supplemental Information 14 Original, uncropped gel for Fig. 6.

Click here for additional data file.

We thank Aulikki Lehmus and Marianne Kuuslahti for the skillful technical assistance with most experiments; Leena Mäkinen, and Hannaleena Piippo, for their technical assistance with zebrafish experiments, and Jukka Lehtiniemi for the artwork of Fig. 14. Thanks are due to Alma Yrjänäinen and Linda Urbański for the help with immunohistochemistry experiments and collecting tissues. We thank Mataleena Parikka for the help with adult zebrafish tissue collection. The authors thank Ritva Romppanen for preparing samples for mass spectrometry analysis. We acknowledge Biocenter Finland for infrastructure support in light scattering experiments. Core facilities at BioMediTech and Faculty of Medicine and Life Sciences, University of Tampere, were essential in microscopy (Tampere Imaging Facility), zebrafish experiments (Zebrafish Laboratory), and in DNA sequencing (Sequencing Facility).

Additional Information and Declarations

Competing Interests

Author Contributions

Animal Ethics

DNA Deposition

Data Availability

The authors declare that they have no competing interests. Seppo Parkkila and Vesa Hytönen were part-time employees of Fimlab Laboratories Ltd. (laboratory center owned by the local university hospital district) at the time of performing the first experiments of this study. Ashok Aspatwar was an employee of Fimlab Laboratories Ltd. for some periods during this study. Csaba Ortutay is an employee of Hiducator Ltd., Prajwol Manandhar is an employee of The Center for Molecular Dynamics Nepal, and Mika Hilvo is an employee of Zora Biosciences Ltd.

Maarit S. Patrikainen conceived and designed the experiments, performed the experiments, analyzed the data, wrote the paper, prepared figures and/or tables, reviewed drafts of the paper.

Martti E.E. Tolvanen conceived and designed the experiments, performed the experiments, analyzed the data, wrote the paper, prepared figures and/or tables, reviewed drafts of the paper.

Ashok Aspatwar conceived and designed the experiments, performed the experiments, analyzed the data, wrote the paper, prepared figures and/or tables, reviewed drafts of the paper.

Harlan R. Barker conceived and designed the experiments, performed the experiments, analyzed the data, wrote the paper, prepared figures and/or tables, reviewed drafts of the paper.

Csaba Ortutay conceived and designed the experiments, performed the experiments, analyzed the data, wrote the paper, prepared figures and/or tables, reviewed drafts of the paper.

Janne Jänis conceived and designed the experiments, performed the experiments, analyzed the data, wrote the paper, prepared figures and/or tables, reviewed drafts of the paper.

Mikko Laitaoja conceived and designed the experiments, performed the experiments, analyzed the data, prepared figures and/or tables, reviewed drafts of the paper.

Vesa P. Hytönen conceived and designed the experiments, performed the experiments, analyzed the data, wrote the paper, prepared figures and/or tables, reviewed drafts of the paper.

Latifeh Azizi performed the experiments, analyzed the data, prepared figures and/or tables, reviewed drafts of the paper.

Prajwol Manandhar performed the experiments, analyzed the data, prepared figures and/or tables, reviewed drafts of the paper.

Edit Jáger performed the experiments, analyzed the data, reviewed drafts of the paper.

Daniela Vullo performed the experiments, analyzed the data, prepared figures and/or tables, reviewed drafts of the paper.

Sampo Kukkurainen performed the experiments, reviewed drafts of the paper, original discovery of CA VI–PTX.

Mika Hilvo analyzed the data, reviewed drafts of the paper, original idea that the PTX might actually pentamerize non-mammalian CA VI.

Claudiu T. Supuran conceived and designed the experiments, analyzed the data, contributed reagents/materials/analysis tools, prepared figures and/or tables, reviewed drafts of the paper.

Seppo Parkkila conceived and designed the experiments, contributed reagents/materials/analysis tools, wrote the paper, reviewed drafts of the paper.

The following information was supplied relating to ethical approvals (i.e., approving body and any reference numbers):

Zebrafish housing and care in the Zebrafish facility of the University of Tampere have been approved by the National Animal Experiment Board of Finland, administered through the Provincial Government of Western Finland, Province Social and Health Department Tampere Regional Service Unit (permit # LSLH-2007-7254/Ym-23). Using five day old zebrafish as a model organism requires no specific ethical permission, likewise studying tissues after euthanizing adult fish.

The following information was supplied regarding the deposition of DNA sequences:

The zebrafish ca6 cDNA sequence has been deposited to the ENA database (http://www.ebi.ac.uk/ena) and assigned the identifier LT724251. The translated protein sequence is available in UniProt (http://www.uniprot.org) as A0A1R4AHH7.

The following information was supplied regarding data availability:

Multiple sequence alignments for phylogenetic trees are provided as files Data S1–S6, and are referenced in the appropriate locations in the text. A multi-page protein sequence alignment is provided as Fig. S1.

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
