# Peer review of "Identification and characterization of a novel zebrafish (Danio rerio) pentraxin–carbonic anhydrase"

_PeerJ, doi:10.7717/peerj.4128_

## Round 0.1 · original submission · Minor Revisions

Dear Martii,

Your manuscript entitled "Identification and characterisation of a novel zebrafish (Danio rerio) pentraxin-carbonic anhydrase" has now been seen by 3 referees. You will see from their comments that they find your work of interest, and that really minor points, easily addressable are raised.

We therefore invite you to revise and resubmit your manuscript as soon as you can, taking into account these minor points raised.

We thank you for choosing PeerJ, as we are committed to providing a fair and constructive peer-review process.

Reviewer 1 ·

Basic reporting

The manuscript is written properly with minor misspellings and other mistakes, like activates (line 94), was taken (?)(127), no comma (135), of (165), contains (lines 514-515), and more.
The literature seems correct and well covered.
The structure of the manuscript is proper with a few logical flaws, at least from my point of view. For instance, why the "The region after the CA domain...." chapter does not directly follow the "Exon lengths..." chapter?
In many instances there are abbreviations used that are not explained, e.g. dpf, MO, hpf that can be difficult for scientists from other fields.
Overall the manuscript is interesting, produces a lot of data and partially answers the question of the function of the pentraxin domain in CA VI.

Experimental design

Experimentally this manuscript meets all criteria needed to be published in PeerJ.
The research question is well defined and it clearly aims at solving the presence of an additional domain in non-mammalian CA VI.
Experiments are well performed, however some data is missing:
1. In sequence searches there are no e-values and scores given. We don't know if there are more distant sequences that could be added to the analysis.
2. Why the sequences with only 20 aa insertions were eliminated from the analysis? Was anything else wrong with them?
3. What was the ground on which you decided to edit the longer sequences? What, except for the usual sequence length, lead you to such a decision?
4. Since both DNA and protein sequences seem very similar why did you decide to analyze both types of trees? Usually it is used for more distant cases.
5. In the manuscript you discuss almost the same frequently figures from the main manuscript and the supplemental ones. I think you should decide about that because it is quite inconvenient for the readers.
6. In the "Evaluation of potential..." part of the Methods you state the presence of a hypothetical 18 residues helix. When reading from the beginning the reader has no idea what it is and why you even mention it.
7. At the end of the "Construction of recombinant..." method you mention another PCR. Honestly, I couldn't get what this PCR is for and what is the difference from the previous one. It should be clarified.
8. In "Quantitative real-time PCR" you do not mention how many times you tried this experiment, for statistical reasons.
9. "In the "Non-mammalian CA VI..." Results section you do give details about conservation of the CA VI amino acids (xx out of yyy), but you miss it for the PTX domain.
10. In the phylogeny figures you do not mention if the sequence used is DNA or protein. It should be mentioned in the figure legend.
11. If you want to emphasize specific data from trees you may colour them accordingly.
12. At the end of the first Results chapter you state that PTX is more similar to short pentraxins, CRPs and SAPs. What is the significance of that? I couldn't really find a conclusive answer to that.
13. In the "Platypus is a candidate..." section you do not mention if you tried to fit this contig into the genome. If so where did it fit the best?
14. In my opinion th e"Light scattering analysis..." chapter should be located before the "Recombinant CA VI-PTX..." chapter.
15. Figure 7 is not too well described. Same for Fig. 8.
16. In Fig. 9 there is an inconsistency. What sequence is right: KGP or KQP...?
17. The difference between figures 9 and S6 are not well defined. It needs a better explanation.
18. Your "Conclusions" is a Summary.

Validity of the findings

This manuscript describes a novel finding and shows interesting data both in theoretical and experimental parts. The results however do not fully answer the question of the specific function of the PTX domain in the CA VI family. The hypothesis of a replaced attaching function is great and needs future analyses.
The results are sound and original and are suitable for publishing after these minor issues are fixed.

Additional comments

I lie this manuscript for its broad theoretical and experimental reach. It would be perfect with "the last thrust" that will prove the binding hypothesis. I hope you get to this point soon.

Reviewer 2 ·

Basic reporting

I enjoyed reading this manuscript. Authors address the interesting case of CA6, which gained a pentraxin domain in the ancestor of vertebrates and then lost it in the ancestor of mammals (eutherians?). This study is very thorough, and puts together high-quality phylogenetics, structural, genomics and functional studies. I recommend its publication.

There are a few questions that could be discussed. I would love to understand why this happened as it did (1st gain, 2nd loss, combination of those two domains). I know it is probably impossible to answer this kind of questions with high confidence, but at least authors could attempt to discuss possible hypotheses on why this domain combination, why not in mammals, what impact it may have had on the respective functions of the gene?

What about CA6 in mammals? A comparison may help understanding the evolutionary history of this gene. Is the loss of the domain associated to variation in sequence conservation patterns, expression of the gene across tissues, etc? According to GTEx, human CA6 is only expressed in skin. Does this provide any hint on the above questions?

It is remarkable that the amphipathic helix has been conserved in mammals (btw, I liked how authors inferred the origin of this helix). This suggests it has an important functional role. Any idea on what this role might be?

Regarding the pentraxin domain, it would be interesting to see if the patterns of conservation / divergence differ compared to other pentraxins. This may give insights on functional divergence of this domain.

In summary, this is a fascinating case for which further functional studies may benefit from attempts to answer the above questions.

Minor comments:
-Introduction, lines 87-90: I think that sentence is not needed
-Methods, lines 168-169: check grammar

Experimental design

Minor comment: the phylogenetic tree of the pentraxin domain would be easier to interpret if paralogous pentraxins from other vertebrates (not only human) were included. This is just a suggestion, it's fine as it is now.

In future follow-up studies authors may want to scan for potential signatures of positive selection associated to these domain gain/loss evolutionary events.

Validity of the findings

no comment

Reviewer 3 ·

Basic reporting

The manuscript by Patrikainen et al., describes the biophysical characterization and bioinformatic analysis (phylogenetic and 3D modelling) of the carbonic anhydrase VI from Danio rerio.

Minor comments:

1- Regarding the 3D modelling of the amphipathic helix. Its 3D localization in the interface between CA and PTX domains is unconvincing (Figure 5).
My recommendation would be to remove it.

2- I consider it appropriate to cite the Pfam database, where
phylogenetic trees and domain architecture can be found, for example:
http://pfam.xfam.org/protein/E9QB97_DANRE

Experimental design

no comment

Validity of the findings

no comment

Additional comments

no comment

External reviews were received for this submission. These reviews were used by the Editor when they made their decision, and can be downloaded below.

---

## Round 0.2 · accepted · Accept

Dear Martti,
I am very happy to inform you that your revised version of your manuscript Identification and characterization of a novel zebrafish (Danio rerio) pentraxin-carbonic anhydrase - has been Accepted for publication. Congratulations!

Best wishes, and thank you for helping us to make a constructive review.

External reviews were received for this submission. These reviews were used by the Editor when they made their decision, and can be downloaded below.